# Trans- and Multigenerational Maternal Social Isolation Stress Programs the Blood Plasma Metabolome in the F3 Generation

**DOI:** 10.3390/metabo12070572

**Published:** 2022-06-22

**Authors:** Joshua P. Heynen, Eric J. Paxman, Prachi Sanghavi, J. Keiko McCreary, Tony Montina, Gerlinde A. S. Metz

**Affiliations:** 1Canadian Centre for Behavioural Neuroscience, Department of Neuroscience, University of Lethbridge, Lethbridge, AB T1K 3M4, Canada; josh.heynen@uleth.ca (J.P.H.); epaxman@ualberta.ca (E.J.P.); sanghavp@myumanitoba.com (P.S.); keiko.mccreary2@uleth.ca (J.K.M.); 2Southern Alberta Genome Sciences Centre, University of Lethbridge, Lethbridge, AB T1K 3M4, Canada; 3Department of Chemistry and Biochemistry, University of Lethbridge, Lethbridge, AB T1K 3M4, Canada

**Keywords:** prenatal maternal stress (PNMS), social isolation stress, metabolomics, transgenerational stress, multigenerational stress, ^1^H NMR spectroscopy, biomarkers, risk prediction, developmental origins of health and disease (DOHaD), diabetes

## Abstract

Metabolic risk factors are among the most common causes of noncommunicable diseases, and stress critically contributes to metabolic risk. In particular, social isolation during pregnancy may represent a salient stressor that affects offspring metabolic health, with potentially adverse consequences for future generations. Here, we used proton nuclear magnetic resonance (^1^H NMR) spectroscopy to analyze the blood plasma metabolomes of the third filial (F3) generation of rats born to lineages that experienced either transgenerational or multigenerational maternal social isolation stress. We show that maternal social isolation induces distinct and robust metabolic profiles in the blood plasma of adult F3 offspring, which are characterized by critical switches in energy metabolism, such as upregulated formate and creatine phosphate metabolisms and downregulated glucose metabolism. Both trans- and multigenerational stress altered plasma metabolomic profiles in adult offspring when compared to controls. Social isolation stress increasingly affected pathways involved in energy metabolism and protein biosynthesis, particularly in branched-chain amino acid synthesis, the tricarboxylic acid cycle (lactate, citrate), muscle performance (alanine, creatine phosphate), and immunoregulation (serine, threonine). Levels of creatine phosphate, leucine, and isoleucine were associated with changes in anxiety-like behaviours in open field exploration. The findings reveal the metabolic underpinnings of epigenetically heritable diseases and suggest that even remote maternal social stress may become a risk factor for metabolic diseases, such as diabetes, and adverse mental health outcomes. Metabolomic signatures of transgenerational stress may aid in the risk prediction and early diagnosis of non-communicable diseases in precision medicine approaches.

## 1. Introduction

The global rates of metabolic diseases have increased sharply over the past few decades [1]. Metabolic diseases in the mother influence the risk of adverse health outcomes in her children, including abnormal brain development, contributing to the high prevalence of impaired emotional and behavioural function and the risk of mental illness [2,3]. Mental illnesses, such as anxiety disorders or major depressive disorders, are the leading causes of disability worldwide [4]. The prevalence of neurodevelopmental disorders and mental illnesses is rapidly growing, increasing in parallel to the rise in metabolic diseases. Currently, the causal relationship between metabolic state in mother, child, and comorbid mental health issues remains unclear [5,6]; however, both conditions are linked to maternal stress and an adverse prenatal environment [7].

Maternal stress affects offspring physiology and neurodevelopment by programming the fetal hypothalamic–pituitary–adrenal (HPA) axis [7,8,9,10]. Fetal HPA axis programming can raise the lifetime risk for altered glucose and insulin metabolism, adiposity [11,12], anxiety disorders, and depression [13]. In addition, maternal anxiety and prenatal stress have been linked to low birth weight [14,15] and the development of metabolic syndromes such as type 2 diabetes, cardiovascular disease, and obesity later in life [16]. Notably, a lack of social support and social isolation represent significant stressors for pregnant women [17]. During the COVID-19 pandemic, perceived social isolation had the largest effects on the substantially elevated symptoms of anxiety (59%) and depression (37%) experienced by pregnant women in Canada [18].

Prenatal stress has been linked to altered DNA methylation marks [19] and microRNA signatures [20,21,22] in both blood and brain tissues. These epigenetic alterations enable the inheritance of metabolic and behavioural phenotypes to subsequent generations [23,24]. Thus, epigenetic markers linked to experience-dependent insulin resistance and diabetes [25,26] and altered affective states [27,28,29] may propagate across generations independent of changes in DNA sequence and affect the aging phenotype [30,31,32]. In fact, the transgenerational phenotype of high anxiety frequently coincides with altered glucose metabolism and other metabolic variations [28,29,30,33,34,35]. Furthermore, work in human cohorts of prenatal maternal stress caused by a natural disaster has shown metabolic alterations that indicate a higher risk of metabolic and cardiovascular disease [36]. Moreover, metabolic risk factors are among the most common causes of noncommunicable diseases [37], and stress critically contributes to metabolic risk [38,39]. Nevertheless, it has been difficult to disentangle the link between stress and metabolic disease risk.

Because upstream epigenetic regulation driven by prenatal or remote ancestral stress is reflected in downstream cellular metabolic functions, the metabolome, i.e., the sum of all metabolites in an organism, arguably represents a direct result of transgenerational programming [34,40,41] and provides insight into the metabolic underpinnings of epigenetically heritable diseases. Here, we used high-resolution proton nuclear magnetic resonance (^1^H NMR) spectroscopy and supervised and unsupervised machine learning approaches to probe robust metabolic signatures in rat blood plasma generated by transgenerational and multigenerational maternal stress in offspring from the third filial generation (F3). The models allow differentiation between the metabolomic impact of transgenerational epigenetic inheritance versus cumulative generational prenatal stress on adverse health outcomes. Both stress models were previously linked to preterm birth and metabolic disease [24], anxiety-like and depression-like symptoms [29,32,35], impaired neurodevelopmental trajectories [42,43], and accelerated biological aging [30,31]. This study reveals clinically accessible peripheral markers that may provide insight into metabolic pathways linked to the programming of adverse health outcomes and altered behaviour based on trans- and multigenerational maternal stress in rats.

## 2. Results

### 2.1. Trans- and Multigenerational Stress Generate Unique Metabolic Profiles

PCA was utilized to identify patterns in a subset of metabolites that were deemed significant using either a Mann–Whitney U test (*p* < 0.05) or VIAVC analysis. Metabolite profiles of the transgenerational prenatal stress (TPS) and multigenerational prenatal stress (MPS) groups were clearly separated from CONT (Figure 1A,B). Despite some overlap between the TPS and MPS profiles compared to CONT, data in Figure 1C demonstrate significant separation between the TPS and MPS groups and minimal confidence interval overlap.

There were 144 total bins created. Of these bins, TPS vs. CONT, MPS vs. CONT, and TPS vs. MPS revealed eight, eight, and thirteen significantly altered bins, respectively. Compared to CONT, both the TPS and MPS groups exhibited upregulated formate and creatine phosphate, as well as downregulated glucose (Table 1). The TPS comparison with the CONT group showed that leucine, isoleucine, alanine, 2-oxoisocaproate, and 2-hydroxyisovalerate were downregulated. Unique to the MPS group, 3-methylxanthine, threonine, and tyramine were upregulated compared to CONT, whereas betaine was downregulated (Table 1). Lastly, a comparison of TPS to MPS showed upregulation of succinate, creatine, and tyramine and downregulation of citrate, lactate, choline, alanine, and serine (Table 1).

### 2.2. Trans- and Multigenerational Stress Differentially Program Amino Acid Metabolism Pathways

Metabolite set enrichment analysis (MSEA) of the TPS compared to the CONT group (Figure 2A) indicated significant differences in valine, leucine, and isoleucine (branch chain amino acids) biosynthesis (*p* < 0.01) and degradation (*p* < 0.01). In addition, the glucose-alanine cycle (*p* < 0.01) and alanine metabolism were altered in the MSEA. Pathway topology supported the effects on branched-chain amino acid synthesis (*p* < 0.01) and degradation (*p* < 0.01) pathways but did not reveal significant pathway hits in the glucose-alanine cycle or alanine metabolic pathways (Figure 2B). The aminoacyl tRNA pathway (*p* < 0.01), however, was also shown to be affected by TPS.

MSEA revealed a significant impact on glycine, serine, and threonine metabolism (*p* = 0.013) in the MPS group when compared to CONT (Figure 3A). The pathway topology analysis reflected this significance, showing changes to the glycine, serine, and threonine metabolic pathway (*p* < 0.01; Figure 3B). No other pathways were found to be altered in this group, although betaine metabolism was somewhat significant in the metabolite set enrichment (*p* = 0.07).

When the impacts of trans- versus multigenerational stress were compared (Figure 4), the metabolite set enrichment analysis found the tricarboxylic acid (TCA) cycle (*p* < 0.01), methionine metabolism (*p* < 0.01), glycine, serine, threonine metabolism (*p* < 0.01), and alanine metabolism (*p* = 0.05) to be significantly affected. The TPS versus MPS topology analysis (Figure 4B) showed differential activation of the glycine, serine, and threonine metabolic pathway (*p* < 0.01) to be most significantly affected, followed by the TCA cycle (*p* < 0.01), the alanine, aspartate, and glutamate metabolic pathway (*p* < 0.01), the cyanoamino acid pathway (*p* = 0.03), the methane metabolic pathway (*p* = 0.05), and the aminoacyl tRNA pathway (*p* = 0.05).

### 2.3. Trans- and Multigenerational Stress-Induced Shifts in Energy Metabolism Are Associated with Altered Exploratory Behaviours

Table 2 presents the Spearman’s rank correlations of creatine phosphate and leucine/isoleucine to open field task metrics. Open field task scores can be found in the Appendix A (Appendix A). These data revealed a significant relationship between vertical exploratory time in the open field task and the concentration of creatine phosphate in the blood plasma metabolome in both TPS (Rho = −0.730, *p* = 0.007) and MPS rats (Rho = −0.691, *p* = 0.013). In the MPS group, creatine phosphate was also significantly correlated with the total distance traveled (Rho = −0.712, *p* = 0.009), the number of vertical moves such as rears (Rho = −0.636, *p* = 0.03), and the distance traveled in the center (Rho = −0.635, *p* = 0.003) of the open field arena (Table 2). In general, less exploratory behaviour was linked to higher creatine phosphate levels.

TPS, when compared to CONT, also displayed significant correlations between leucine/isoleucine and total distance (Rho = 0.646, *p* = 0.02), vertical time (Rho = 0.66, *p* = 0.02), and central distance (Rho = 0.596, *p* = 0.02) in the open field task (Table 2). MPS, when compared to CONT, showed significant correlations between leucine/isoleucine in the blood plasma and the distance traveled in the center of the open field task (Rho = 0.656, *p* = 0.02; Table 2). In general, lower exploratory activity was linked to reduced leucine/isoleucine levels.

## 3. Discussion

Metabolic risk factors are among the most common causes of noncommunicable diseases, and stress critically contributes to metabolic risk. This study shows that ancestral maternal social isolation stress induces distinct and robust metabolic profiles in adulthood, which are characterized by significant changes in energy metabolism, such as upregulated formate and creatine phosphate, and downregulated glucose. The plasma metabolome induced by transgenerational stress differs from recurrent maternal stress in terms of energy and amino acid metabolisms, cellular signaling, and immunoregulation. The findings indicate that maternal stress may lead to long-term metabolic adjustments and risks across generations of offspring, leading to altered tissue function and potentially contributing to an altered behavioural phenotype. The findings suggest that ancestral maternal stress may be a risk factor in idiopathic insulin resistance, diabetes, obesity, and other chronic and non-communicable diseases.

Substantial metabolic changes may explain the significant consequences of maternal stress on neuroendocrine and cardiometabolic functions, brain development, and emotional and cognitive impairments observed in the exposed offspring [10,22,44,45]. Moreover, it was shown that paternal stress can result in transgenerational programming of a metabolic phenotype in the F4 generation of the male lineage [34,41]. Here, we also show that trans- and multigenerational maternal stress causes distinct metabolic profiles in blood plasma, providing potentially predictive signatures of adverse health outcomes in the offspring.

Both TPS and MPS altered 21 metabolites known to be present in blood plasma. Among them, upregulation of creatine phosphate and formate represented a primary marker of ancestral stress. Creatine phosphate plays a vital role in ATP regeneration in skeletal muscle and is a rate-limiting pathway in muscle performance enabling brief, high-power activity [46] or meeting higher energy demand in response to stress [47]. On the other hand, elevated formate may trigger a metabolic switch from low to high adenine nucleotide levels, increasing the rate of glycolysis and intracellular lactate levels [48]. The involvement of these metabolites in both TPS and MPS suggests that ancestral stress may program a permanently upregulated energy metabolism, which supports a link between programmed stress vulnerability and documented low birth weights, lifetime diabetes and obesity risk, and other metabolic abnormalities in offspring exposed to stress in utero [10,49,50].

Metabolic abnormalities at birth are also considered a risk factor for mental disorders such as schizophrenia [51], depression [52], and attention deficit hyperactivity disorder [14,53]. Here we show that ancestral stress alters critical energy metabolism pathways, such as branched-chain amino acid (BCAA) biosynthesis and the tricarboxylic acid cycle, which mediate neurological and behavioural outcomes [54]. The proteinogenic BCCAs leucine, isoleucine, and valine are essential amino acids involved in glucose metabolism, neurotransmitter synthesis, and neuronal function. Thus, disruption in BCAA metabolism has been linked to abnormal brain development [54] and lifetime risk of neurodegenerative disorders, such as Alzheimer’s disease [55]. These mechanisms also implicate BCAAs in the regulation of immune functions, as nutritional interventions of both supplemented and reduced BCAA intake have been shown to affect metabolic and inflammatory functions [56,57].

When interfering and non-significant metabolites were excluded, the unsupervised PCA showed clear class separation between TPS and MPS, indicating metabolic profiles unique to each stress lineage. Interestingly, citrate and lactate were found to be significantly downregulated in the TPS group as opposed to MPS, but neither had a significantly altered citrate level compared to non-stressed controls. Both citrate and lactate are involved in energy generation, particularly in the production of ATP in the tricarboxylic acid (TCA) cycle, and in muscular ATP regeneration. If lactate levels are low in TPS animals, then metabolic acidosis and muscular fatigue may occur more quickly [58]. Similarly, low citrate levels in MPS animals may indicate energy deficiency [59]. By contrast, a relative increase in lactate in MPS animals, indicative of stress hyperlactataemia, is also secondary to anaerobic glycolysis induced by tissue hypoperfusion or hypoxia [60]. While here it may serve adaptive functions, in disease states stress hyperlactataemia is a reliable predictor of mortality [61]. This reflects the phenotype of MPS animals, whose metabolic and mental health impairments [24,31] and higher morbidity [30] exceed those of TPS animals.

TPS rats also revealed significantly downregulated choline and serine levels compared to the MPS group. Choline and its metabolites are required for several physiological functions, such as cell signaling and cholinergic transmission [62], and may be involved in blood lipid regulation and the development of cardiovascular disease and cognitive decline in aging [63]. Serine plays a central role in cellular proliferation and in nervous system development and functioning [64]. Altered levels of serine in patients with psychiatric disorders and neurological abnormalities underscore its importance in brain development and function [65]. Abnormal levels of these metabolites may help to explain anxiety-like behaviours in ancestrally stressed offspring, but it remains unclear why TPS is linked to lower levels than MPS. It has been hypothesized that cumulative stress across generations is mediated by an adaptive response to the constant presence of a stressor [66], which at times may facilitate partial physiological adaptation and resilience to recurrent stress in MPS animals [43,67].

Overall, ancestral stress had the largest impact on protein biosynthesis, energy generation, and immunity. Thus, the formation and breakdown of amino acids, as well as energy metabolism, are likely involved in the physiological and behavioural response to both TPS and MPS. These results are reinforced by the correlational analysis demonstrating associations between particular metabolites with anxiety-like behaviours. Previous studies have associated reduced exploratory behaviour, such as vertical rearing, with higher levels of anxiety [68]. Furthermore, trans- and multigenerationally stressed rats showed higher stress sensitivity and precocious onset of motor hyperactivity and risk assessment behaviours [69]. Here, anxiety-like behaviours, including reduction in the amount of vertical rearing and total and central distance traveled in the open field task, were associated with higher circulating creatine phosphate levels in the MPS group. Notably, the association between increased creatine phosphate and reduced rearing activity (vertical time), which is a behaviour linked to increased anxiety and psychomotor inhibition [70], was observed in both the MPS and TPS groups. However, many of the psychophysiological links to anxiety-like behaviours in the open field task are tenuous [71]. Nevertheless, it remains to be further determined if creatine phosphate concentrations may serve as a marker of anxiety risk in populations directly exposed to or with a family history of prenatal stress.

BCAAs such as leucine and isoleucine provided further insight into the metabolic underpinning of anxiety-like behaviours. More time spent rearing was linked to greater leucine and isoleucine concentrations in rats from the TPS group. Greater levels of leucine and isoleucine were associated with more distance travelled in the center of the open field in the MPS group. This change was mirrored in the TPS lineage with greater levels of leucine and isoleucine translating to greater total and central distance travelled in the open field task, linking remote ancestral stress to altered cellular energy metabolism and behavioural change. Thus, higher levels of leucine and isoleucine may be linked to higher stress resilience, whereas lower levels of these BCAAs may serve as markers of anxiety-like behaviours. Lower levels of BCAAs have also been linked to adverse metabolic and inflammatory functions [56,57] in addition to poorer stroke recovery, increased risk of renal disease, and type 2 diabetes [72,73,74,75]. Causal inferences of BCAA metabolism in brain function and behaviour require further investigation.

The present downstream cellular metabolic changes may reflect differential underlying epigenetic changes that are potentially heritable. The TPS generation of the maternal lineage in particular excludes direct exposure effects and unambiguously suggests heritable epigenetic transgenerational mechanisms for offspring three generations removed from maternal stress [76,77] to regulate cellular metabolism. Epigenetics may mediate the relationship between genotype and internal and external environments [33] and cellular metabolic changes may directly reflect changes in these upstream regulatory pathways. Furthermore, gestational stress may impact the developing child through the placenta or through direct or indirect effects on the fetal brain, potentially contributing to the differences observed between TPS and MPS cohorts. However, the current findings must be interpreted carefully, as this study only involved males and only included a small number of control animals.

The present study used a novel approach to quantify outcomes of ancestral stress and HPA axis dysregulation by taking a metabolomic perspective and providing possible mechanisms for health outcomes in an animal model. The metabolomic biomarkers identified in this study may provide potential biomarkers for the risk prediction and early diagnosis of non-genetically heritable stress-related diseases. The identification of predictive and diagnostic metabolic biomarkers that identify adverse health trajectories linked to maternal stress is critical for precision medicine approaches, enabling the identification of the most vulnerable individuals who are at the highest risk. Because the orchestrated regulation of metabolic pathways is essential for stress regulation, development, and successful aging, its failure may lead to cell and tissue dysfunction and result in chronic and non-communicable diseases. On the other hand, metabolic profiles are often cell-type specific [78], thus providing an opportunity for the discovery of causal pathogenic mechanisms and early diagnosis of a disease. Due to the potentially significant health impact of social isolation, understanding the metabolic consequences of ancestral stress may provide new personalized therapeutic strategies to mitigate stress-associated adverse health outcomes.

## 4. Materials and Methods

### 4.1. Experimental Design

#### 4.1.1. Animal Model

The study involved 24 adult male Long-Evans rats obtained from the F3 generation of three lineages: yoked controls (CONT, *n* = 8), transgenerational prenatal stress (TPS; *n* = 8), and multigenerational prenatal stress (MPS; *n* = 8). Four samples from the control group did not yield enough plasma for metabolomics analysis, resulting in a reduced control group size (CONT, *n* = 4). TPS rats were the third filial (F3) generation of a lineage in which only the first filial (F1) generation was stressed prenatally; subsequent generations remained unstressed (Figure 5). MPS rats were the F3 generation of a lineage in which each consecutive generation (F1, F2 and F3) was prenatally stressed.

#### 4.1.2. Gestational Stress

Pregnant dams were stressed by social isolation, which has been shown to result in mild psychosocial stress in rats [79] and profound epigenetic and behavioural phenotypes [69,80]. Rats were housed in pairs from weaning until postnatal (P) day 90. Then, dams were separated and housed alone, one per cage, throughout pregnancy until weaning of their offspring [69,80]. Pairing for mating began at P90 and the maximum time spent in social isolation prior to conception was 30 days. Control rats remained housed in pairs, except for the period from gestational day 21 to lactational day 21 (time of weaning), to allow for undisturbed rearing of their offspring.

#### 4.1.3. Breeding Colony

Animals were bred in-house for at least five generations prior to the beginning of the experiment. Each generation F0-F4 was outcrossed to avoid inbreeding. Distinct lineages were monitored through the JAX Colony Management System (JCMS; Jackson Laboratory, Bar Harbour, ME, USA). One male offspring per litter was randomly selected for this metabolomics study. Bystander effects of stress were avoided by using designated testing and housing spaces. All housing, handling, testing, and tissue sampling conditions were harmonized and carefully controlled across generations.

The rats were housed in polycarbonate shoebox cages on corn cob bedding under a 12 h light/dark cycle with lights on at 7:30 AM. The room temperature was maintained at 20 °C with relative humidity at 30%. All procedures were performed in accordance with the guidelines of the Canadian Council on Animal Care and approved by the University of Lethbridge Animal Welfare Committee.

### 4.2. Behavioural Testing

Anxiety-like behaviours towards a novel environment and risk taking were assessed in an open field test at P90. Rats were individually placed into an Accuscan activity monitoring box (Accuscan Instruments Inc., Columbus, OH, USA; 42 × 42 × 30 cm) and recorded for 10 min. VersaMax™ software (Accuscan Instruments Inc., OH, USA) monitored the animals’ horizontal and vertical activity to be analyzed using VersaDat™ software (Accuscan Instruments Inc., Columbus, OH, USA; [29]). Vertical (rearing activity) and horizontal (distance travelled) exploration were considered.

### 4.3. Sample Collection and Preparation

At the age of 140 days (P140), 600 μL of blood was collected from the lateral tail vein using a 23-gauge butterfly needle coated with lithium heparin, while rats were anaesthetized using 4% isoflurane. All blood samples were collected at the same time of day, between 8:00 and 9:00 AM in the morning. EDTA and citrate collection tubes were avoided, as they give additional spectral signals in ^1^H NMR spectroscopy [81]. Blood was transferred to centrifuge tubes and plasma was obtained by centrifugation at 1600× *g* for 15 min at 4 °C. Samples were stored at −80 °C until further processing.

Water soluble metabolites were extracted from plasma using a methanol precipitation protocol [82]. Methanol was added in a 2:1 ratio (550 μL buffered plasma + 1.10 mL methanol) in 2.0 mL Eppendorf tubes. These were vortexed for 5 s, incubated at −20 °C for 20 min, and centrifuged at 12,000× *g* for 30 min at 4 °C. Supernatant was decanted to fresh tubes and allowed to dry under a gentle stream of nitrogen gas until all liquid had evaporated. Dried pellets were then re-suspended in 600 μL of phosphate buffer calibrated to pH 7.4 and vortexed for 10 s, or until the pellets were completely dissolved. The phosphate buffer was prepared as a 4:1 ratio of KH2PO4:K2HPO4 in a 4:1 H2O:D2O solution to obtain a final concentration of 0.5 M. The D2O contained 0.03% *w*/*v* trimethylsilyl propanoic acid (TSP) as a chemical shift reference for ^1^H NMR spectroscopy. To maintain metabolite integrity, 0.03% *w*/*v* of sodium azide (NaN3) was also added to the buffer as an antimicrobial agent. Finally, samples were centrifuged again at 12,000× *g* for 5 min at 4 °C and 550 μL of re-suspended supernatant was added to 5 mm NMR tubes for data acquisition.

### 4.4. NMR Data Acquisition and Processing

NMR spectra were collected on a 700 MHz Bruker Avance III HD spectrometer (Bruker, Milton, ON, Canada; Appendix A). The 1-D NOESY gradient water suppression pulse sequence ‘noesygpr1d’ was used. Each sample was run for 512 scans to a total acquisition size of 256 k. The spectra were zero filled to 512 k, automatically phased and baseline corrected, and line-broadened by 0.3 Hz. The processed spectra were then exported to MATLAB (The MathWorks, Natick, MA, USA) for statistical analysis. Spectra were binned using Dynamic Adaptive Binning [83]. Each spectrum was normalized to the total unit area of all spectral bins and the spectral region corresponding to water was removed from data normalization. The data set was then Pareto scaled to reduce the influence of intense peaks while emphasizing weaker ones and log transformed [84]. All peaks were referenced to TSP (0.00δ).

### 4.5. Statistical Analysis

The overall structure of scaled spectra was visualized and compared across control and experimental groups using principal component analysis (PCA). A Shapiro–Wilk test for data normality was applied and all data were found to be non-parametric. A Mann–Whitney U test was subsequently used to determine which spectral bins were significantly altered in comparison groups [85]. Bonferroni–Holm correction was applied to all univariate statistical tests to correct for multiple comparisons. Bins with *p*-values less than or equal to 0.05 were deemed significant and considered as potential markers defining substantial alterations in metabolite concentration between groups.

PCA and Mann–Whitney U tests were carried out using the web-based Metaboanalyst software [86] and MATLAB^®^ (MathWorks, Natick, MA, USA), respectively. The variable importance analysis based on random variable combination (VIAVC) algorithm [87] was utilized to determine which set of metabolite bins led to the best discrimination between the comparison groups (referred to as the best subset). This machine learning, multivariate statistical algorithm makes use of random data combinations and iteratively tests each combination using both PLS-DA and the area under the curve of the receiver operator characteristic curve (ROC) to determine the best subset of metabolites leading to class separation [88]. PCA score plots (Figure 1) were generated using only those bins identified as significant by Mann–Whitey U and VIAVC testing (Table 1). Pathway topology analysis and metabolite set enrichment analysis were carried out using Metaboanalyst [89,90]. The Rattus Norvegicus pathway library was utilized for both analyses, the hypergeometric test was selected for the over-representation analysis, and relative-betweenness centrality was chosen for the pathway topology analysis.

Spearman’s rank correlations were carried out using SPSS 26 for Windows (IBM Corporation, Armonk, NY, USA) to determine the relationship between normalized metabolite concentrations and behavioural data. This analysis concerned only metabolites determined to be significantly altered between groups, based on the multivariate and univariate statistical tests outlined above. Samples with *p*-values equal to or less than 0.05 were considered significant.

### 4.6. Metabolite Identification

A spectral database of pure metabolites was generated and used to identify most metabolites found in the NMR spectra. The Human Metabolome Database [91,92,93] was used to supplement and aid the creation of our spectral library, and to identify some substances that were not obtained for the creation of this database. Furthermore, tools of the Chenomx 8.2 NMR Suite (Chenomx Inc., Edmonton, AB, Canada) were used for identifying and quantifying NMR metabolites which allowed for the spectral deconvolution of biofluid samples into individual components.

## Figures and Tables

**Figure 1 metabolites-12-00572-f001:**
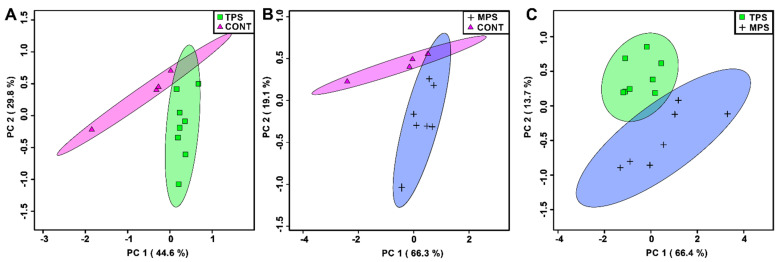
Principle component analysis (PCA) plots showing unsupervised separation between the groups: (**A**) transgenerationally stressed rats (TPS) vs. non-stressed controls (CONT); (**B**) multigenerationally stressed rats (MPS) vs. non-stressed controls (CONT); and (**C**) transgenerationally stressed rats (TPS) vs. multigenerationally stressed rats (MPS). Each PCA was carried out using only the bins that were determined to be significantly altered. Each point (triangle, cross, circle, or square) represents one individual based on the list of blood plasma metabolites found to be statistically significant via a Mann–Whitney U test and VIAVC analysis. Coloured ellipses represent 95% confidence intervals. *x* and *y* axes show principal components 1 and 2, respectively, with brackets indicating the variance explained by each principal component. Note that the metabolomes of both TPS and MPS differed substantially from CONT and from each other.

**Figure 2 metabolites-12-00572-f002:**
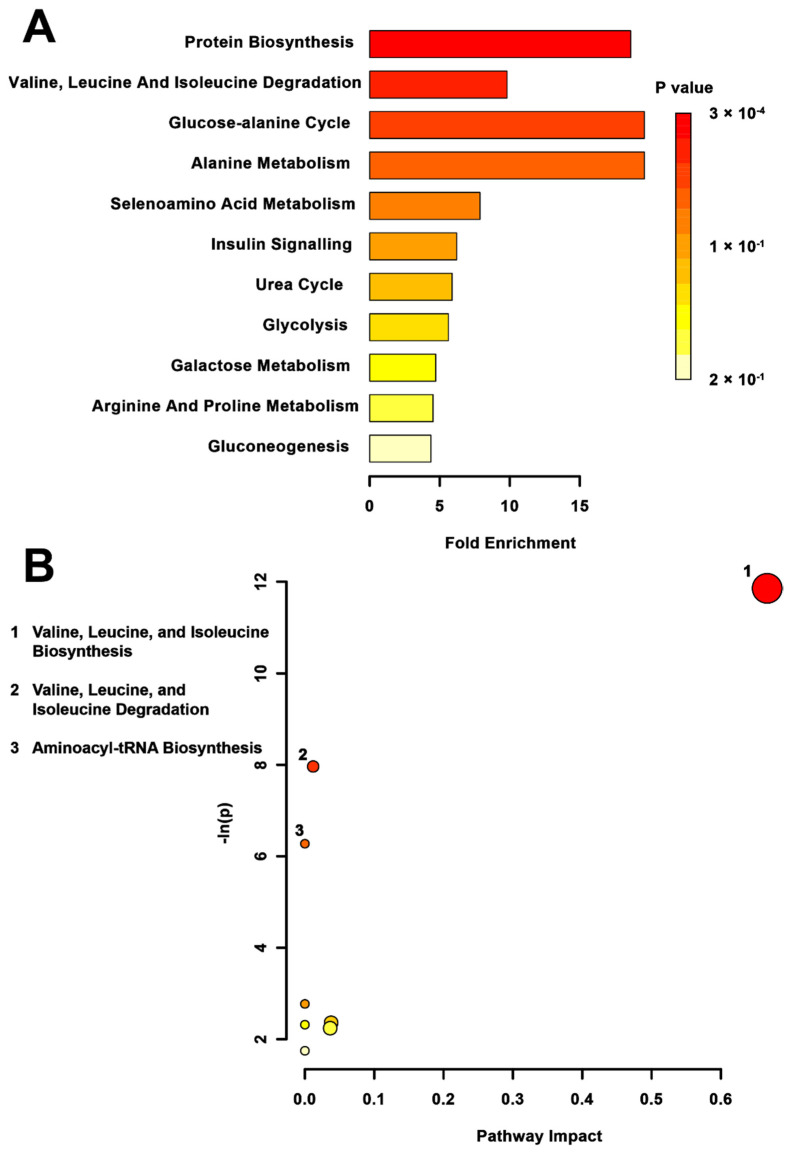
Pathway-associated metabolite set enrichment analysis (MSEA) for TPS vs. CONT. (**A**) Plot showing affected biological processes in transgenerationally stressed (TPS) compared to non-stressed (CONT) rats based on metabolites identified as significantly altered between these groups. The *p*-value for each pathway is shown using the heatmap on the right of the figure and the fold enrichment shows how many times greater than chance the process is involved. (**B**) Metabolomic pathway analysis showing all matched pathways according to *p*-values from pathway enrichment analysis and pathway impact values from pathway topology analysis in TPS and non-stressed CONT rats. The *y*-axis shows the negative natural log of *p*, such that a higher value on the *y*-axis gives a lower *p*-value. The *x*-axis gives the pathway impact, which correlates to the number of metabolite hits in a particular pathway. Only metabolic pathways with *p* < 0.05 are labeled. TPS had the highest impact on valine, leucin, and isoleucine biosynthesis.

**Figure 3 metabolites-12-00572-f003:**
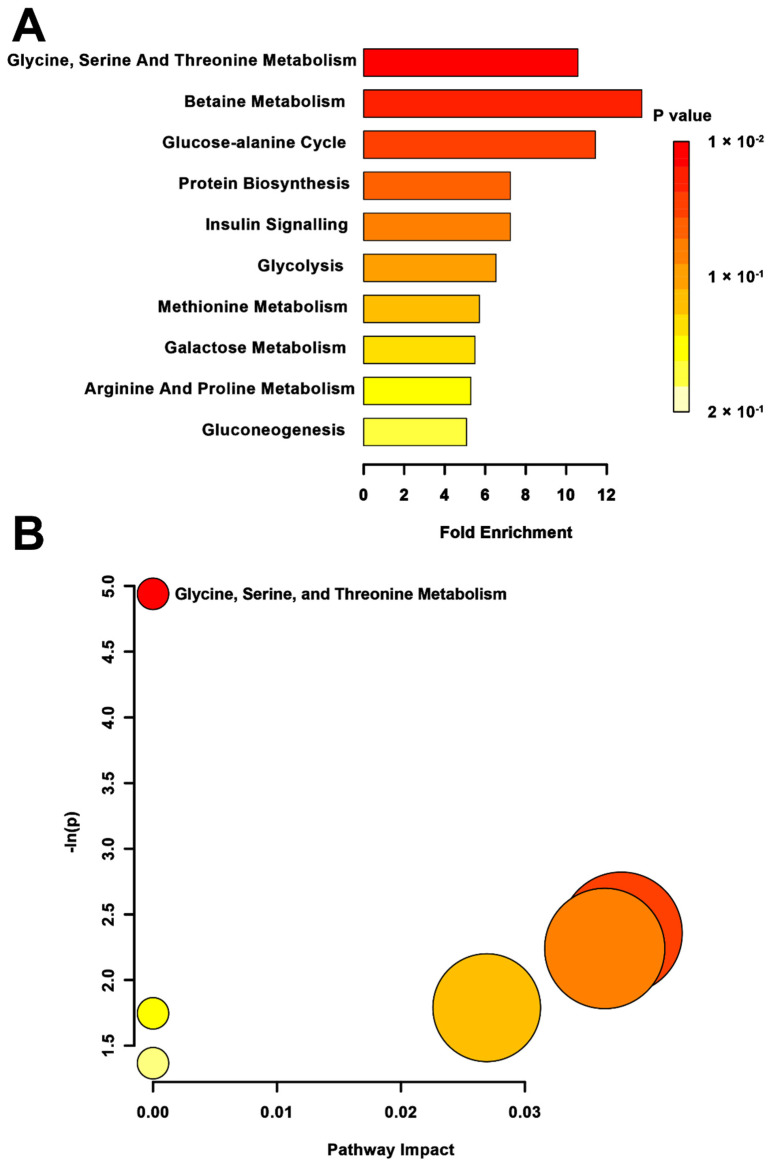
Pathway-associated metabolite set enrichment analysis (MSEA) for MPS vs. CONT. (**A**) Plot displaying affected biological processes in multigenerationally stressed (MPS) compared to non-stressed control (CONT) rats based on metabolites identified as significantly different between these groups. The *p*-value for each pathway is indicated in the gradient on the right side. The fold enrichment shows how many times greater than chance the pathway is involved. (**B**) Metabolomic pathway analysis showing all matched pathways according to *p*-values from pathway enrichment analysis and pathway impact values from pathway topology analysis in MPS and non-stressed CONT rats. The *y*-axis shows the negative natural log of *p*, such that a higher value on the *y*-axis gives a lower *p*-value. The *x*-axis indicates the pathway impact, which correlates to the number of metabolite hits in a particular pathway. Only metabolic pathways with *p* < 0.05 are labeled. MPS had the highest impact on the glycine, serine, and threonine metabolism.

**Figure 4 metabolites-12-00572-f004:**
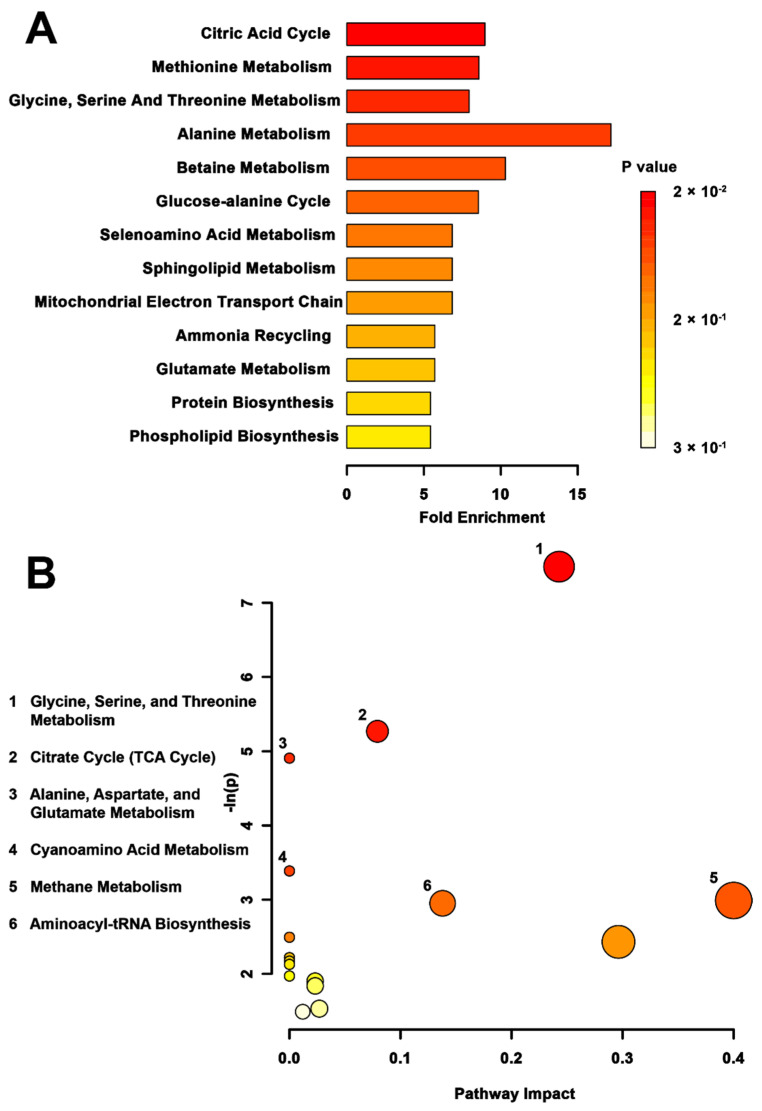
Pathway-associated metabolite set enrichment analysis (MSEA) for TPS vs. MPS. (**A**) Plot showing affected biological processes in transgenerationally stressed (TPS) compared to multigenerationally stressed (MPS) rats, based on metabolites identified as significantly altered between these groups. The *p*-value for each pathway is shown using the heatmap on the right of the figure and the fold enrichment shows how many times greater than chance the process is involved. (**B**) Metabolomic pathway analysis showing all matched pathways according to *p*-values from pathway enrichment analysis and pathway impact values from pathway topology analysis in TPS vs. MPS rats. The *y*-axis shows the negative natural log of *p*, such that a higher value on the *y*-axis gives a lower *p*-value. The *x*-axis indicates the pathway impact, which correlates to the number of metabolite hits in a particular pathway. Only metabolic pathways with *p* < 0.05 are labeled. The largest difference in pathway activity between TPS and MPS was observed in the glycine, serine, and threonine metabolism.

**Figure 5 metabolites-12-00572-f005:**
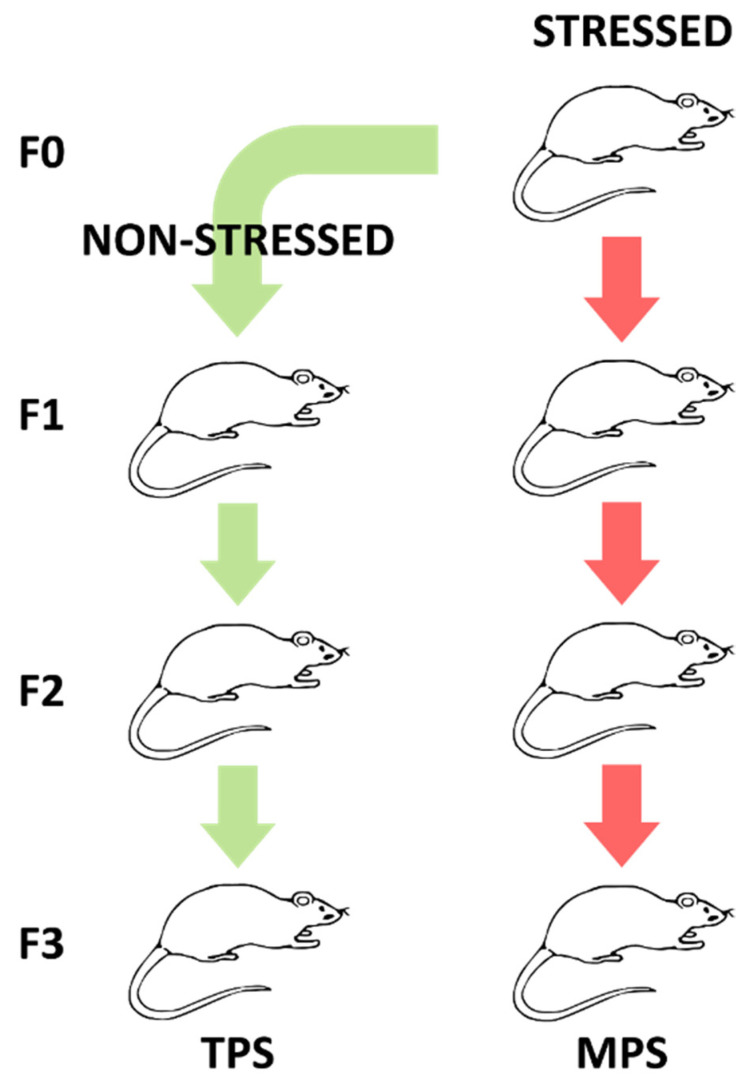
Illustration summarizing the experimental design generating lineages of transgenerational prenatal stress (TPS) and multigenerational prenatal stress (MPS) through filial generation zero (F0) to F3.

**Table 1 metabolites-12-00572-t001:** *p*-values of blood plasma metabolites found to be significant in male Long-Evans rats in either a Mann–Whitney U test, the variable importance analysis based on random variable combination (VIAVC), or both. Rats were either non-stressed (CONT), transgenerationally-stressed (TPS), or multigenerationally-stressed (MPS). Up- or downregulation of the metabolites is also indicated. Metabolites with multiple peaks are represented as metabolite.1, metabolite.2… metabolite.n.

Group	Metabolite	NMR Chemical Shift Range of Bin (ppm)	Mann-Whitney U Test	VIAVC	VIP Score	Regulation
F3-TPS vs. F3-CONT	Creatine phosphate	3.035641–3.028	1.62 × 10^−2^	4.40 × 10^−9^	1.51	Up
	Formate	8.5343705–8.442	2.83 × 10^−2^	2.09 × 10^−6^	1.51	Up
	Glucose.1	3.715–3.704	1.09 × 10^−1^	7.02 × 10^−6^	0.87	Down
	Leucine.1, isoleucine.1, 2-hydroxyisovalerate.1	0.9586–0.9476	7.27 × 10^−2^	1.12 × 10^−47^	0.84	Down
	Alanine	1.498–1.4878	4.85 × 10^−2^	2.35 × 10^−20^	0.81	Down
	Glucose.2	3.526–3.514123	3.68 × 10^−1^	1.20 × 10^−7^	0.80	Down
	Leucine.2, isoleucine.2	0.9812075–0.9682	4.85 × 10^−2^	1.85 × 10^−30^	0.72	Down
	2-Hydroxyisovalerate.2, 2-oxoisocaproate	0.9476–0.9133615	5.70 × 10^−1^	6.83 × 10^−22^	0.40	Down
F3-MPS vs. F3-CONT	Singlet at 8.38 ppm	8.442–8.388584	1.09 × 10^−1^	3.46 × 10^−37^	1.54	Up
	Formate	8.5343705–8.442	7.27 × 10^−2^	3.03 × 10^−54^	1.14	Up
	Creatine phosphate	3.035641–3.028	2.83 × 10^−2^	2.39 × 10^−2^	1.09	Up
	3-Methylxanthine	8.388584–8.0675875	3.68 × 10^−1^	4.02 × 10^−31^	1.02	Up
	Threonine	3.5998105–3.59	4.85 × 10^−2^	1.66 × 10^−20^	0.94	Up
	Tyramine.1	7.0614405–6.05	6.83 × 10^−1^	1.25 × 10^−20^	0.82	Up
	Glucose, betaine	3.28–3.2684065	7.27 × 10^−2^	2.81 × 10^−31^	0.64	Down
	Tyramine.2	7.2946365–7.0614405	9.33 × 10^−1^	3.90 × 10^−27^	0.35	Up
F3-TPS vs. F3-MPS	Citrate.1	2.68–2.6728	6.50 × 10^−2^	5.94 × 10^−11^	1.64	Down
	Citrate.2	2.543–2.5269	8.30 × 10^−2^	3.34 × 10^−12^	1.40	Down
	Citrate.3	2.5269135–2.511	1.05 × 10^−1^	7.66 × 10^−11^	1.31	Down
	Citrate.4	2.6727785–2.62297	1.05 × 10^−1^	2.25 × 10^−8^	1.17	Down
	Choline	3.22–3.203	2.07 × 10^−2^	1.75 × 10^−13^	0.91	Down
	Lactate	4.137413–4.12681	3.79 × 10^−2^	-	0.91	Down
	Succinate	2.412–2.402	1.95 × 10^−1^	1.01 × 10^−10^	0.89	Up
	Alanine.1	1.487783–1.476	8.30 × 10^−2^	9.29 × 10^−16^	0.89	Down
	Alanine.2	1.498–1.4878	2.34 × 10^−1^	3.88 × 10^−10^	0.83	Down
	Serine.1, creatine	3.971–3.937	2.34 × 10^−1^	4.95 × 10^−8^	0.77	Up
	Serine.2	3.8517095–3.849	4.99 × 10^−2^	5.3 × 10^−3^	0.65	Down
	Alanine.3	3.8066745–3.798575	1.61 × 10^−1^	1.27 × 10^−8^	0.45	Down
	Tyramine	7.2946365–7.0614405	2.34 × 10^−1^	2.99 × 10^−8^	0.36	Up

**Table 2 metabolites-12-00572-t002:** Spearman’s rank correlations of blood plasma metabolite concentrations of creatine phosphate and leucine/isoleucine in relation to open field (OF) behavioural measurements. Correlation coefficients were obtained utilizing metabolite concentrations from comparisons of non-stressed (CONT) versus multigenerationally-stressed (MPS) Long-Evans rats and non-stressed (CONT) versus transgenerationally-stressed (TPS) Long-Evans rats. *—indicates a correlation with a *p*-value less than 0.05; **—indicates a correlation with a p-value less than 0.01.

	Creatine Phosphate				Leucine/Isoleucine			
Test	TPS vs. CONT		MPS vs. CONT		TPS vs. CONT		MPS vs. CONT	
	Rho	*p*-value	Rho	*p*-value	Rho	*p*-value	Rho	*p*-value
Total Distance			−0.712 **	0.009	0.646 *	0.023		
Number of Vertical Moves			−0.636 *	0.026				
Vertical Time	−0.730 **	0.007	−0.691 *	0.013	0.660 *	0.02		
Central Distance			−0.635 *	0.026	0.674 *	0.016	0.656 *	0.02

## Data Availability

The data presented in this study are available upon request from the corresponding authors, as it has not been uploaded to an online database.

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
