# Peer review of "Trans- and Multigenerational Maternal Social Isolation Stress Programs the Blood Plasma Metabolome in the F3 Generation"

_metabolites, 2022, doi:10.3390/metabo12070572_

Round 1

Reviewer 1 Report

Introduction

In the introduction, the use of affirmative form e.g. “Health impacts of prenatal stress are linked to altered DNA methylation (ref)…” is frequent, and is too strong. The formulation like “it has been shown a link between prenatal stress and DNA methylation”, should be preferred, or use the modal verbs “may, can” etc..

Methods

N=4 for control group is low and the experiment should have been repeated to reach at least n=8 like in other groups.

Statistics

- Since there are 3 groups, an ANOVA should be performed before the pair-wise comparisons. The Kruskall-wallis ANOVA would be appropriate since this is a non parametric test. It should be followed by the Man-Whitney test with correction for multiple comparisons, since with n=3, there are potentially 6 pair-wise comparisons which can be performed. Some software include this correction automatically.

- In addition, the pair-wise comparison should also be followed by a FDR correction, e.g. the benjamini-hochberg test, since the number of variables can be very high with buckets and the risk of false positive increase a lot with the number of variables.

- The number of buckets should be provided.

- It is not clear if PCA were conducted with all bins or only with significant buckets, or both. In figure 1 it seems that PCA were conducted with significant variables only. In that case, the 3 PCA represented figure 1 did not use the same number of variables?

Since PLS-DA are produced using The VIAC algorithm, they could be shown, may be as sup figure, with their quality factors assessment.

- A figure with 1H-NMR spectra of plasma extract and assignment must be provided, perhaps as supplementary data

- Was the blood sampled at the same period in the day, e.g. all in the morning? Were the animals submitted to a 2 hours fasting before sampling, in order to avoid the impact of food intake on blood metabolome?

Results

We observe in the score plot of the PCA (fig 1) that among the control group, one animal appears very different from the 3 others, these latter being closer to TPS or MPS animals. This points out the necessity to have additional control animals. As a consequence, the comparisons TPS/CONT and MPS/CONT appear as questionable.

Author Response

The reviewers raised very thoughtful suggestions that allowed us to significantly improve the quality of the manuscript. We appreciate the considerable time investment of carefully reading the manuscript and providing these helpful suggestions. We have addressed the specific concerns as outlined in the following.

Reviewer #1

Comment 1: Introduction

In the introduction, the use of affirmative form e.g. “Health impacts of prenatal stress are linked to altered DNA methylation (ref)…” is frequent, and is too strong. The formulation like “it has been shown a link between prenatal stress and DNA methylation”, should be preferred, or use the modal verbs “may, can” etc…

Response:

Thank you for the comment, we have adjusted this section according to your suggestion. The sentence now reads: “Prenatal stress has been linked to altered DNA methylation marks [19] and microRNA signatures [20-22].”

We have also reviewed the rest of the manuscript and made the appropriate adjustments with respect to your suggestion.

Comment 2: Methods

N=4 for control group is low and the experiment should have been repeated to reach at least n=8 like in other groups.

Response:

We regret the small sample size and acknowledge that represents a limitation to our study. The study of transgenerational inheritance of a phenotype under closely controlled conditions is very challenging. Only one male offspring per litter was selected for this study to avoid confounding effects. We have clarified this procedure on page 12 in the section “Breeding Colony”. Because parallel studies were performed, a limited sample size for this metabolomics study was available. On the other hand, the fact that many big data approaches in genomics, such as microRNA analyses, have successfully used n=4 or even n=3 to identify robust predictive or diagnostic biomarker signatures gave us confidence to go ahead with this sample size. Partial Least Squares -Discriminant Analysis (PLS-DA) is a supervised method for determining separation between comparison groups. We have used two different approaches to confirm the significance of the supervised results, according to Westerhuis et al. (2008) and Szymanska et al. (2012): 1000-fold permutation testing and double 10-fold cross validation testing. This approach provided a solid approach to metabolite identification. Furthermore, given that metabolomics as well identifies a signature rather than a single biomarker provides a robust result, as reflected by the significant findings in the present study.

Comment 3: Statistics

Since there are 3 groups, an ANOVA should be preformed before the pair-wise comparisons. The Kruskall-wallis ANOVA would be appropriate since this is a non parametric test. It should be followed by the Man-Whitney test with correction for multiple comparisons, since n=3, there are potentially 6 pair-wise comparisons which can be preformed. Some software include this correction automatically.

In addition, the pair-wise comparison should also be followed by a FDR correction, e.g. the benjamini-hochberg test, since the number of variables can be very high with buckets and the risk of false positive increase a lot with the number of variables.

Response:

Thank you for your comment. We utilized a Shapiro Wilk test to check for data normality and subsequently applied a Mann-Whitney U test, as the data was determined to be non-parametric. This is explained in lines 420-423 of the methods section where we state the following:

“A Shapiro Wilk test for data normality was applied and all data was found to be non-parametric. A Mann-Whitney U test was subsequently used to determine which spectral bins were significantly altered in comparison groups [51].”

In the case of this work, we utilized the decision tree algorithm outlined by the Goodpaster et. al. paper entitled “Statistical significance analysis of nuclear magnetic resonance based metabonomic data” (2010). This paper is reference 51 indicated in the text above.

As this study is looking at how differences in ancestral stress alter the plasma metabolome, we are only interested in comparing each experimental group to the controls, and then with one another. We then utilized the results of the Mann-Whitney U test to determine which bins were significantly altered. In the field of metabolomics there are several different ways in which univariate tests can be applied. Some groups prefer the use of ANOVA while others prefer the use of Mann-Whitney U tests or the students t-test, depending on the data normality. Our choice of statistical methods is one of the standard ways to carry out this analysis in metabolomics and is justified by the reference mentioned above.

With respect to correcting our tests for multiple comparisons, the reviewer is correct that this needs to be done. All measurements presented in this manuscript were Bonferroni-Holm corrected, as specified in the above-mentioned reference (51). This method of correction is far more stringent than correcting based on false discovery rate. We thank the reviewer for catching this missing piece of methods information and we have added the following to the methods section to clarify this:

“Bonferroni-Holm correction was applied to all univariate statistical tests to correct for multiple comparisons.”

Comment 4:

The number of buckets should be provided.

Response:

We have addressed your comment and added a sentence on line 107-108 in the results section that states the total number of bins that were created, as well as the total number of significant bins for each comparison.

“There were 144 total bins created. Of these bins, TPS vs CONT, MPS vs CONT, and TPS vs MPS revealed eight, eight and thirteen significantly altered bins, respectively.”

Comment 5:

It is not clear if PCA were conducted with all bins or only with significant buckets, or both. In figure 1 it seems that PCA were conducted with significant variables only. In that case, the 3 PCA represented figure 1 did not use the same number of variables?

Response:

Thank you for your comment. We addressed this issue on lines 262-264 as follows: “When interfering and non-significant metabolites were excluded, the unsupervised PCA showed clear class separation between TPS and MPS, indicating metabolic profiles unique to each stress lineage.”

However, the reviewer is correct that this required further clarification in the manuscript. The principal component analyses (PCA) were conducted using significant bins only. The number of bins and the corresponding identified metabolites are provided in Table 1. Furthermore, text clarifying the number of significant bins was added to lines 107-108 in response to comment 4 from the reviewer above. We have also added the following sentence to lines 440-441 of the methods section to add further clarity that the PCA’s were constructed using only the significant variables:

“PCA score plots (Figure 1) were generated using only those bins identified as significant by Mann-Whitey U and VIAVC testing (Table 1).”

In addition, to make the manuscript easier to read and follow, we have also added the following sentence to the caption of figure 1: “Each PCA was carried out using only the bins that were determined to be significantly altered”

Comment 6:

Since PLS-DA are produced using The VIAC algorithm, they could be shown, may be as sub figure, with their quality factors assessment.

Response:

Although VIAVC utilizes PLS-DA, it does not generate PLS-DA scores plots, as VIAVC utilizes PLS-DA to create the model of bins for use in determining variable importance via the percent increase or decrease of the area under the receiver operator characteristic curve. Including the PLS-DA scores plots generated by the VIAVC algorithm would mean including hundreds of figures, as VIAVC iteratively produces as many models as needed to include and exclude each variable.

Comment 7:

A figure with 1H-NMR spectra of plasma extract and assignment must be provided, perhaps as supplementary data

Response:

The reviewer is correct that this should be included in the manuscript. We have included a figure of the spectrum from the rat plasma with the significant metabolites labelled in the supplemental data as Figure S1.

Comment 8:

Was the blood sampled at the same period in the day, e.g. all in the morning? Were the animals submitted to a 2 hours fasting before sampling, in order to avoid the impact of food intake on blood metabolome?

Response:

Thank you for pointing out the need to clarify the sample collection times. All samples were collected at the same time of day, between 8:00-9:00 AM in the morning. On page 12, we have included a statement in the methods section to clarify the sample collection times. In our facility, lights go on at 7:00 AM. Considering that rats are a nocturnal species with a fast metabolism, the animals likely had a fasting interval of at least 1 hour prior to blood sample collection. All animals had ad libitum access to the same foods to remove the effect of diet on metabolome variability.

Comment 9: Results

We observe in the score plot of the PCA (fig 1) that among the control group, one animal appears very different from the 3 others, these latter being closer to TPS or MPS animals. This points out the necessity to have additional control animals. As a consequence, the comparisons TPS/CONT and MPS/CONT appear as questionable.

Response:

Despite the normalization of data, figure 1 displays PCA plots which are unsupervised scores plots. As such, the variability among animals is retained, leading to one CONT animal appearing to have a dissimilar metabolic profile than other CONT animals. Unfortunately, due to circumstances mentioned in the paper and in response to comment 2 above, 4 plasma samples from the CONT group were not able to be processed. We regret the small sample size and acknowledge that this limits the reliability of the figures. To make this point clear in the paper, we have added a sentence at the end of the paper on lines 329-331.

“However, the current findings must be interpreted carefully, as this study only involved males and only included a small number of control animals.”

Reviewer 2 Report

In this study the authors utilize proton nuclear magnetic resonance spectroscopy to analyze blood plasma metabolome from F3 rats born from lineages that experience a single, transgenerational, stressor or multiple generations of stress. Stress consisted of social isolation stress. This was modeled by housing pregnant dams alone during the perinatal, gestational and lactational, window while control animals remained pair house throughout the duration of the experiment. While the topic and study design are of significant interest and relevance to recent mental health concerns surrounding the pandemic, the descriptive nature of the results, lack of sex-specific comparisons, and limited brain/behaioral assessments reduces the overall impact of these findings for the DOHaD field. Additionally, metabolomics at other developmental timepoints, not just in adulthood would have been advantageous - changes in these metabolites in adulthood may not be the same during the neonatal and peripubertal windows when the brain is still developing. However, the authors did find some interesting similarities and differences in metabolic profiles between TPS and MPS that may aid in risk prediction/biomarker identification of non-communicable diseases.  

  1. Maybe I’m misinterpreting the text at line 96, but the confidence interval ellipses for TPS and MPS groups do overlap, not by much, but it doesn’t quite seem like you can make the claim that they do not overlap. 

  2. What was the justification for only looking at male offspring? Many studies have shown that programmatic effects of stress often manifest in a sex-specific manner

  3. Methods state that pregnant dams were housed alone throughout pregnancy until weaning. Can the authors provide more detail about when during gestation, i.e. gestational day, they were singly housed? Was this when they were found to have a copulation plug or was there another metric used to identify and separate pregnant females?

  4. Why tail vein blood and not cardiac puncture? Seems like that would have been beneficial given that some collections did not provide enough blood for analysis.

  5. Line 362 you mention that a maximum of 3 offspring per litter were selected for experiments. Was this taken into consideration in statistical analyses? I didn’t see any mention of using litter as a covariate during data analysis.

  6. Line 362 also says per litter of each sex, so were both males and females used? Or just males? If both sexes were used please consider running sex-specific analyses. If not please adjust the current text because right now it is not clear.

  7. Anxiety testing was performed by I do not see any figures or results that provide details of what was found in the open field test. At least in supplement. 

  8. I see at the bottom of the manuscript there is reference to a supplementary figure and table. Might be worth referencing these somewhere in the text if they are relevant to discussed findings. For example, if one of these includes behavioral findings.  

  9. This is very minor but I think it makes more sense to switch the order in which Statistical Analysis and Metabolite Identification are presented in the methods section. 

  10. Was body weight, length, etc considered at all? Seems like a relevant phenotypic outcome that should have been taken into consideration.

  11. I think it would be good to add some discussion points on limitations of this study, including sex and lack of thorough assessments of animal behavior and body morphometrics. Additionally, the authors should discuss that stress has been found to impact the placenta and developing brain in utero which may speak to differences found between TPS and MPS cohorts. In other words, programming is multifaceted and complex. The reported metabolic changes are only part of the picture.

Author Response

The reviewers raised very thoughtful suggestions that allowed us to significantly improve the quality of the manuscript. We appreciate the considerable time investment of carefully reading the manuscript and providing these helpful suggestions. We have addressed the specific concerns as outlined in the following.

Reviewer #2

In this study the authors utilize proton nuclear magnetic resonance spectroscopy to analyze blood plasma metabolome from F3 rats born from lineages that experience a single, transgenerational, stressor or multiple generations of stress. Stress consisted of social isolation stress. This was modeled by housing pregnant dams alone during the perinatal, gestational and lactational, window while control animals remained pair house throughout the duration of the experiment. While the topic and study design are of significant interest and relevance to recent mental health concerns surrounding the pandemic, the descriptive nature of the results, lack of sex-specific comparisons, and limited brain/behavioral assessments reduces the overall impact of these finding for the DOHaD field. Additionally, metabolomics at other developmental timepoints, not just in adulthood may not be the same during the neonatal and peripubertal windows when the brain is still developing. However, the authors did find some interesting similarities and differences in metabolic profiles between TPS and MPS that may aid in risk prediction/biomarker identification of non-communicable diseases.

Comment 1:

Maybe I’m misinterpreting the text at line 96, but the confidence interval ellipses for TPS and MPS groups do overlap, not by much, but it doesn’t quite seem like you can make the claim that they do not overlap.

Response:

The reviewer is correct, thank you for identifying that oversight. We have adjusted the sentence on lines 95-97.

“Despite some overlap between TPS and MPS profiles compared to CONT, data in Figure 1C demonstrate significant separation between TPS and MPS groups and minimal confidence interval overlap.”

Comment 2:

What was the justification for only looking at male offspring? Many studies have shown that programmatic effects of stress often manifest in a sex-specific manner.

Response:

Thank you for raising this very critical issue. This study is part of a larger framework in which we investigated male and female rats.

We regret that for this metabolomics study we were not able to include both males and females. The study of transgenerational inheritance of a phenotype under closely controlled conditions is very challenging. Our parallel publications do highlight the importance of studying sex differences, however, this first exploratory study of NMR spectroscopy aimed to determine if the trans- and intergenerational epigenetic and transcriptomic changes we observed will result in any metabolic differences. Thus, this study was to prove that profound cellular metabolic function is affected by transgenerational stress in just male rats. Future follow-up studies will include female offspring as well to identify potential sex differences.

Comment 3:

Methods state that pregnant dams were housed alone throughout pregnancy until weaning. Can the authors provide more detail about when during gestation, i.e. gestational day, they were singly housed? Was this when they were found to have a copulation plug or was there another metric used to identify and separate pregnant females?

Response:

According to our earlier work (McCreary et al., 2016a, 2016b), female rats were singly housed starting on postnatal day 90 throughout pregnancy until weaning of their offspring. Pairing for mating began at P90 and the maximum time spent in social isolation prior to conception was 30 days. Control rats were housed in pairs until gestational day 21. Thus, our “stress” groups underwent preconception social isolation followed by gestational isolation.

We have addressed this comment in the manuscript on line 365-396: “Pairing for mating began at P90 and the maximum time spent in social isolation prior to conception was 30 days. Control rats remained housed in pairs, except for the period from gestational day 21 to lactational day 21 (time of weaning), to allow for undisturbed rearing of their offspring.”

Please see also:

McCreary, J.K.; Erickson, Z.T.; Metz, G.A. Environmental enrichment mitigates the impact of ancestral stress on motor skill and corticospinal tract plasticity. Neurosci. Lett. 2016, 632, 181-186, doi:10.1016/j.neulet.2016.08.059.

McCreary, J.K.; Erickson, Z.T.; Hao, Y.; Ilnytskyy, Y.; Kovalchuk, I.; Metz, G.A. Environmental Intervention as a Therapy for Adverse Programming by Ancestral Stress. Sci Rep 2016, 6, 37814, doi:10.1038/srep37814.

Comment 4:

Why tail vein blood and not cardiac puncture? Seems like that would have been beneficial given that some collections did not provide enough blood for analysis.

Response:

Tail vein puncture was utilized as the animals in this study were used for further breeding and continued research. The investigation of psychological stress by social isolation prevented us from using more invasive procedures to collect blood samples, such as cardiac puncture. The blood collection in these animals was not done at terminal end point, hence we have attempted to minimize stress in the animals as far as possible.

Comment 5:

Line 362 you mention that a maximum of 3 offspring per litter were selected for experiments. Was this taken into consideration in statistical analyses? I didn’t see any mention of using litter as a covariate during data analysis.

Response:

We would like to thank the reviewer for their insightful comment. We did not use an ANCOVA, and therefore did not control for litter as a covariate. However, disregarding the control animal with variability, we observed clustering of the metabolome from animals in different litters with unsupervised PCA testing. This strongly supports that there is no experimental repetition effect based on the litter.

Comment 6:

Line 362 also says per litter of each sex, so were both males and females used? Or just males? If both sexes were used please consider running sex-specific analyses. If not please adjust the current text because right now it is not clear.

Response:

The original line 362 referred to the experimental details of the breeding colony and as such both sexes are included. We have now revised this section to avoid misleading statements. The metabolomics analysis of trans- and multigenerational stress was only carried out on males, which is indicated where the authors state the following:

“The study involved 24 adult male Long-Evans rats obtained from the F3 generation of three lineages”

In addition, we have added the following text to clarify the statement in the methods that the reviewer has specifically pointed out (now line 377):

“One male offspring per litter was randomly selected for this metabolomics study.”

Comment 7:

Anxiety testing was performed by I do not see any figures or results that provide details of what was found in the open field test. At least in supplement.

Response:

Thank you for your suggestion. We have created supplemental table 1 (Table S1) which details the results of the open field test.

Comment 8:

I see at the bottom of the manuscript there is a reference to a supplementary figure and table. Might be worth referencing these somewhere in the text if they are relevant to discussed findings. For example, if one of these includes behavioral findings.

Response:

Thank you for your comment. We have added in a reference to the open field task measures as supplemental table 1 (Table S1) on lines 194-195.

“Open field task scores can be found in the supplemental material (Table S1).”

We have also added a figure of the NMR spectrum as supplemental figure 1 in response to a suggestion from a different reviewer (line 414).

“NMR spectra were collected on a 700 MHz Bruker Avance III HD spectrometer (Bruker, ON, Canada; Figure S1).”

Comment 9:

This is very minor but I think it makes more sense to switch the order in which Statistical Analysis and Metabolite Identification are presented in the methods section.

Response:

Thank you for comment. We believe that the Metabolite Identification section should remain after the Statistical Analysis section, as metabolites are identified only after we determine which bins are significant through statistical analysis.

Comment 10:

Was body weight, length, etc considered at all? Seems like a relevant phenotypic outcome that should have been taken into consideration.

Response:

Body weight was considered in spearman correlation analysis, however, no significant correlations or a rho value above ±0.6 were observed. However, in response to your comment we have added body weight measures at postnatal day 90 to supplemental table 1 (Table S1).

Comment 11:

I think it would be good to add some discussion points on limitations of this study, including sex and lack of thorough assessments of animal behaviour and body morphometrics. Additionally, the authors should discuss that stress has been found to impact the placenta and developing brain in utero which may speak to differences found between TPS and MPS cohorts. In other words, programming is multifaceted and complex. The reported metabolic changes are only part of the picture.

Response:

 Thank you for your insights and thoughtful suggestions. We acknowledge that the metabolic profiles we utilize for this study indeed may be reflective of upstream alterations which may be due to changes in the placenta or the developing brain. To address these comments and the limitations of our study we have added additional statements to the Discussion section, including on lines 327-331, which read as follows:

“Furthermore, gestational stress may impact the developing child through the placenta or through direct or indirect effects on the fetal brain, potentially contributing to the differences observed between TPS and MPS cohorts.”

“However, the current findings must be interpreted carefully, as this study only utilized males and only included a small number of control animals.”

We did not refer to animal behaviour or body morphometrics in the limitations section but instead have included supplemental table 1 to address the concerns by providing additional behavioural and body metrics.

Reviewer 3 Report

Congratulations to the authors on a timely and interesting manuscript describing metabolic changes in a rat model of trans- and multi-generational stress. My main concern is with the material presented in Table 2 and claims about altered behavioral phenotype. Namely:

Table 2: Are the unreported correlation coefficients and p-values not shown because they are insignificant? Unless there is justification for omitting these data, please show them. I am also not clear on the column headings TPS vs CONT, MPS vs CONT. If I understand correctly, for instance rho=-0.712/p=0.009, you are calculating the correlation between total distance and creatine phosphate in MPS animals. So where is the CONT group in the calculation? Is there a correlation between creatine phosphate and total distance in CONT animals? Is it weaker than in MPS/TPS animals? Please explain this section better and consider reorganizing the table.

In the discussion, you state that “The findings indicate that maternal stress may lead to long-term metabolic adjustments and risks in generations offspring, leading to altered tissue function and potentially contributing to an altered behavioural phenotype.” This is a powerful statement, but it may be unwarranted since the authors do not present summary data describing behavioral tests in each group. I would like to see mean/standard deviation and p-values for total distance, number of vertical moves, vertical time, and central distance comparing the 3 groups. 

Otherwise, I have some minor comments:

The abstract clearly states that the study was conducted in rats but this key information is missing from the introduction. 

Separation between the experimental groups in PCA (Figure 1) is really quite impressive. However, I recommend changing “no confidence interval overlap” in line 96 to “minimal confidence interval overlap”.

I strongly recommend adding a conclusion in section 5. 

Author Response

The reviewers raised very thoughtful suggestions that allowed us to significantly improve the quality of the manuscript. We appreciate the considerable time investment of carefully reading the manuscript and providing these helpful suggestions. We have addressed the specific concerns as outlined in the following.

Reviewer #3

Comment 1:

Congratulations to the authors on a timely and interesting manuscript describing metabolic changes in a rat model of trans- and multi-generational stress. My main concern in with the material presented in Table 2 and claims about altered behavioral phenotype. Namely:

Table 2: Are the unreported correlation coefficients and p-values not shown because they are insignificant? Unless there is justification for omitting these data, please show them. I am also not clear on the column headings TPS vs CONT, MPS vs CONT. If I understand correctly, for instance rho=-0.712/p0.009, you are calculating the correlation between total distance and creatine phosphate in MPS animals. So where is the CONT group in the calculation? Is there a correlation between creatine phosphate and total distance in CONT animals? Is it weaker than in MPS/TPS animals? Please explain this section better and consider reorganizing the table.

Response:

Thank you for your comment. If the reviewer is referring to the correlation coefficients and p-values comparing open field test measures to metabolites other than those in table 2, we did not include those results as they were either not significant or the rho values were not above ±0.6. Regarding the second part of the reviewer’s comment, we determine significant bins by comparing the concentration of bins between comparison groups; namely, transgenerational stress versus controls; multigenerational stress versus controls; and transgenerational versus multigenerational stress. In order to determine the relationship between a bin and an open field test measure, we correlate differences in bin concentrations between comparison groups and the change in open field test scores between comparison groups. In other words, the control animals and experimental group are both plotted to show that there is a true correlation between a change in metabolite concentration across the two groups. Showing only one of the groups would not illustrate this correlation and would only serve to indicate if there is a difference or a correlation of that metabolite within the experimental group. Furthermore, as bin significance is determined by comparing MPS/TPS to CONT or one another, it is important to label the headings as they are, as those bins may only be significant within that comparison. Finally, we could calculate the correlation between creatine phosphate and CONT animals, however, the relationship between the two would have less meaning without understanding how creatine phosphate concentrations and open field test scores change between comparison groups (as addressed above).

Comment 2:

In the discussion, you state that “The findings indicate that maternal stress may lead to long-term metabolic adjustments and risks in generations offspring, leading to altered tissue function and potentially contributing to an altered behavioural phenotype.” This is a powerful statement, but it may be unwarranted since the authors do not present summary data describing behavioural tests in each group. I would like to see mean/standard deviation and p-values for total distance, number of vertical moves, vertical time, and central distance comparing the 3 groups.

Response:

We thank the reviewer for pointing out this lack of support for the statement we make. We have now included the behavioural test outcomes, as well as the means and standard deviations, as supplemental table 1. We have also referenced supplemental table 1 in the results section on lines 194-195. With respect to p-values, the results of this study correlate these behavioural measurements with significantly altered metabolites and the p-value and rhos for each of the significant correlations is provided in the text.

Comment 3:

Otherwise, I have some minor comments:

The abstract clearly states that the study was conducted in rats but this key information is missing from the introduction.

Response:

We have addressed your comment and made changes to sentences on lines 77-81 and 85-88 to clarify.

“Here, we used high-resolution proton nuclear magnetic resonance (1H NMR) spectroscopy and supervised and unsupervised machine learning approaches to probe robust metabolic signatures in rat blood plasma generated by transgenerational and multigenerational maternal stress in offspring from the third filial generation (F3).”

“This study reveals clinically accessible peripheral markers to provide insight into metabolic pathways linked to programming of adverse health outcomes based on trans- and multigenerational maternal stress in rats.”

Comment 4:

Separation between the experimental groups in PCA (Figure 1) is really quite impressive. However, I recommend changing “no confidence interval overlap” in line 96 to “minimal confidence interval overlap”.

Response:

Thank you for the comment, we have adjusted this section according to your suggestion. The following changes were made on lines 95-97.

“Despite some overlap between TPS and MPS profiles compared to CONT, data in Figure 1C demonstrate significant separation between TPS and MPS groups and minimal confidence interval overlap.”

Comment 5:

I strongly recommend adding a conclusion in section 5.

Response:

We thank the reviewer for this comment. The authors feel that the summary provided at the end of the discussion section (lines 320-345) present a summary of the key findings of the study. For this reason, the authors have chosen to not include the “conclusion section” as it would be redundant with the end of the discussion section. We have removed the reference to section 5 (Conclusions) from the manuscript to add further clarity.

Round 2

Reviewer 1 Report

The authors have addressed the main concerns I had about the introduction, and about the methods, mainly statistics, and the manuscript has been significantly improved.

My main concern was about the low number of animals in the control group, which weakens the results. The authors claim in their response that “…..the fact that many big data approaches in genomics, such as microRNA analyses, have successfully used n=4 or even n=3 to identify robust predictive or diagnostic biomarker signatures gave us confidence to go ahead with this sample size”, and I do not totally agree with this. The metabolome is highly variable, as it is impacted by lots of exogenous and endogenous parameters, even in lab animals. Moreover, metabolites are often produced by several different pathways, and as a result, metabolite changes are most of the time small, since not all the pathways are dysregulated. This is why their variations are not expressed in fold changes like with genes and RNA. As a consequence I think that n=4 or 3 is not at all sufficient in metabolomics study.

However, by considering the fact that “… the study of transgenerational inherence of a phenotype is very challenging”, as claimed by the authors, and the added sentence “However, the current findings must be interpreted carefully, as this study only involved males and only included a small number of control animals”, I consider that my concern has been correctly addressed.
